# An online training platform for SPECT imaging technology utilizing three-dimensional modeling

**Lihua Qiao**[1☯], **Hongzhi Wang**[2☯], **Xiaorui Guo**[3], **Xinkun Lei**[3], **Hua Shang**[3], **Ruibin Zhao**[1], **Tian Xia**[4], **Ruiping Qin**[1], **Zikun Fang**[1], **Luqi Shou**[1], **Yiwen Qin**[1], **Dandan Shang**[5¤*]

**1** School of Medical Imaging, Hebei Medical University, Shijiazhuang, Hebei, China, **2** Shanghai Key Laboratory of Magnetic Resonance, East China Normal University, Shanghai, China, **3** Department of Nuclear Medicine, the Second Affiliated Hospital of Hebei Medical University, Shijiazhuang, Hebei, China, **4** Shanghai Peiyun Education Technology Company Limited, Shanghai, China, **5** Biochemistry and molecular biology, Hebei Medical University, Shijiazhuang, Hebei, China

☯ These authors contributed equally to this work.
¤ Current Address: Biochemistry and molecular biology, Hebei Medical University, Shijiazhuang, Hebei Province, China.
* lily.dandan@163.com

## Abstract

### Objective

The provision of nuclear medicine experimental classes within universities poses significant challenges due to the high risk, substantial cost, and large size of the requisite equipment. To address the bespoke training needs of students majoring in imaging technology in this new medical era, our team has developed an online training platform specifically for SPECT imaging technology, a key aspect of nuclear medicine.

### Method

This platform utilises advanced technologies such as Unity3D to create six three-dimensional scene modules of SPECT imaging and seven typical disease examination operation processes. As a result, we have achieved a human-computer interactive three-dimensional virtual system. The aforementioned teaching strategy has been implemented in our institution's instructional practice across five semesters, in addition to being adopted by three other medical colleges.

### Results

Training data reveals that the overall sample score distribution aligns with a normal distribution, suggesting that the platform's structure is logically and effectively designed. Furthermore, the linear fit slopes of individual sample scores are consistently positive, indicating that the frequency of training sessions yields a positive feedback effect on students' bespoke training. The innovative nature of this platform is protected through computer software copyrights.

**Data availability statement:** All relevant data are within the manuscript and its Supporting Information files.

**Funding:** This research was facilitated through the generous support of the following projects:Research and Practice on Innovation and Entrepreneurship Education and Teaching Reform in Universities of Hebei Province in 2023 (2023cxcy068), Medical Science Research Project of Hebei Province in 2022 (20220971), Research and Practice on Higher Education Teaching Reform in Hebei Province (2022GJJG149), Educational and Teaching Research Project of Hebei Medical University (2022YBZD-4, 2022YBPT-8, 2024CHYB-48), and University Student Innovation Experiment Plan Project of Hebei Medical University in 2023 (USIP2023337).But, The funders had no role in study design, data collection and analysis, decision to publish, or preparation of the manuscript. The author(s) received no specific funding for this work.

**Competing interests:** The authors have declared that no competing interests exist.

## Conclusion

Our online training platform enhances course structure and student training objectives, effectively accommodating the requirements of nuclear medicine teaching for personalized student training, innovative thinking, and the "early clinical" mindset.

## Introduction

Single Photon Emission Computed Tomography (SPECT) is a crucial technique in nuclear medicine examinations, significantly impacting the diagnosis and treatment of various human diseases. As the training requirements for students specializing in medical imaging techniques and clinical technicians in emerging medical fields become increasingly stringent, the importance of implementing teaching experiments to familiarize students with SPECT examinations, encourage clinical thinking, and prepare them for future roles as clinical technicians is profound. However, the radioactivity of nuclear medicine drugs, coupled with the high cost, intensive consumption, and irreversible nature of these experiments, pose significant challenges to conducting nuclear medicine experiments [1]. Consequently, many medical schools do not include these experiments in their curriculum. In order to effectively address this pedagogical challenge, a departure from conventional teaching methodologies is imperative. However,VR has clear benefits such as ease of controlling repetition, feedback, and motivation, as well as overall advantages in safety, time, space, equipment, cost efficiency, and ease of documentation [2]. The development of a distributed medical imaging virtual simulation platform has the potential to steer the application of VR towards a more scientifically rigorous, digitally advanced, and information-rich direction [3].

This educational reform is rooted in the "student-centered" pedagogical approach, emphasizing the principles of "solidifying foundations, adequate practice, respecting individuality, and seeking truth and practicality" to implement innovative teaching designs. Our team developed a three-dimensional model educational training platform using Unity3D technology for SPECT virtual experiments. This platform hosts three-dimensional scene models and process models, facilitating human-computer interaction functions for seven distinct examination projects. Students can thus operate with a comprehensive understanding of the principles and technical know-how. The entire experimental process encompasses learning principles, understanding equipment, operating examination procedures, and diagnosing typical diseases. This enables students to acquire SPECT-related knowledge in a more coherent and systematic manner. The integration of research feedback into teaching presents an innovative strategy for nuclear medicine education, addressing the personalized training needs of students.

## Methods

### Content reconstruction

The process of reconstructing teaching content effectively converts the initial modular teaching approach into a chain teaching methodology. This integration amalgamates

knowledge from the fields of medicine, engineering, and science, thereby consolidating theoretical understanding and reinforcing foundational principles. SPECT is an integral component of nuclear medicine within the realm of medical imaging principles. The subject matter of nuclear medicine is typically taught in a modular fashion across various related courses. For instance, a course in medical physics elucidates the physical foundations of nuclear medicine, while a course on medical imaging principles explores its imaging mechanisms. Similarly, courses on medical imaging equipment, examination technology, image processing, and diagnostics delve into their respective areas within the field of nuclear medicine. Traditional pedagogy presents nuclear medicine technology across several related modular courses, employing what has been termed "modular teaching." This pedagogical reformation employs a virtual experimental platform to transition from modular to chain teaching. In this approach, knowledge pertaining to nuclear medicine is presented in a sequential manner within the imaging principle course. While the primary focus remains on the imaging principles of nuclear medicine, the course also provides brief overviews of its corresponding physical bases, as well as detailed descriptions of relevant imaging equipment, examination technology, image processing techniques, and typical medical case diagnoses. Consequently, individual knowledge points of nuclear medicine imaging technology are interconnected in a sequential chain-like manner. This facilitates a comprehensive reconstruction and integration of the teaching content. This method enables students to more systematically master the theoretical aspects of the subject matter. The reconstruction schematic is shown in Fig 1.

## Platform construction

Medical virtual simulation experiments leverage virtual reality software development tools to replicate abstract and highly practical medical experimental content within a virtual simulation system [4–6]. Over the past decades, several authors

| Curriculum \ Content | X-ray imaging | nuclear medicine imaging | magnetic resonance imaging | ultrasound imaging |
|---|---|---|---|---|
| Medical Physics | The physical basis of X-rays | The physical basis of nuclear | The physical basis of nuclear magnetic resonance | The physical basis of ultrasound |
| Principles of Medical Imaging | The principle of X-ray imaging | The imaging principles of nuclear medicine | The principle of nuclear magnetic resonance imaging | The imaging principle of ultrasound |
| Medical imaging equipment | X-ray equipment | Nuclear medicine equipment | Nuclear magnetic resonance equipment | ultrasonic equipment |
| Medical imaging examination technology | Radiological protection and inspection techniques | Nuclear medicine examination techniques | Magnetic resonance technique | Ultrasonic examination technique |
| Medical Image Processing | X-ray image reconstruction | Nuclear medicine image reconstruction | Nuclear magnetic image reconstruction | Ultrasonic image reconstruction |
| Medical Imaging Diagnostics | X-ray CT image diagnosis | Nuclear medicine image diagnosis | Magnetic resonance image diagnosis | Ultrasonic image diagnosis |

(Horizontal modularization; Longitudinal chaining)

**Fig 1. Content reconstruction diagram.** Horizontal teaching refers to a modular approach, whereas vertical teaching alludes to a chain methodology.

[7–10] have championed and implemented virtual experiments, thereby accelerating the educational process. The use of medical virtual simulation experiments in medical education compensates for the shortcomings of experimental teaching conditions, overcomes the limitations of time and space, and circumvents the various risks associated with physical experimental operations [5,6]. Consequently, the rapid development of virtual simulation experiments [11–19] has been observed across diverse fields such as X-CT [20–23], anatomy [24–26], and dentistry [27].

For the aforementioned reasons, our team has pioneered the development of a nuclear medicine (SPECT) examination technology simulation software, equipped with AI assistant functionalities. Notably, this system is the first of its kind, both domestically and internationally. This innovative tool effectively bridges the gap between theoretical knowledge and practical application. The sophisticated three-dimensional scene rendering offers an engaging virtual environment for students, while the human-computer interactive operations enable them to emulate the roles of nuclear medicine technologists. Furthermore, the integrated AI assistant function facilitates an autonomous learning teaching mode. This flexible platform empowers students to practice at their own pace, free from constraints related to time, location, or frequency.

This model is primarily designed with three key aspects in mind: data acquisition and processing prerequisites, the computer simulation of human-computer interaction within a three-dimensional virtual system, and feedback optimization. The conceptual design framework for this model is depicted in Fig 2.

The conceptual design diagram illustrates the theoretical foundation as the cornerstone, computer numerical simulation as the technical means, and clinical experimental data as both the support and verification. By integrating computer software and hardware, a three-dimensional system of human-computer interaction is achieved. The formative evaluation of the operator can be relayed to the background, thereby optimizing the system model further.

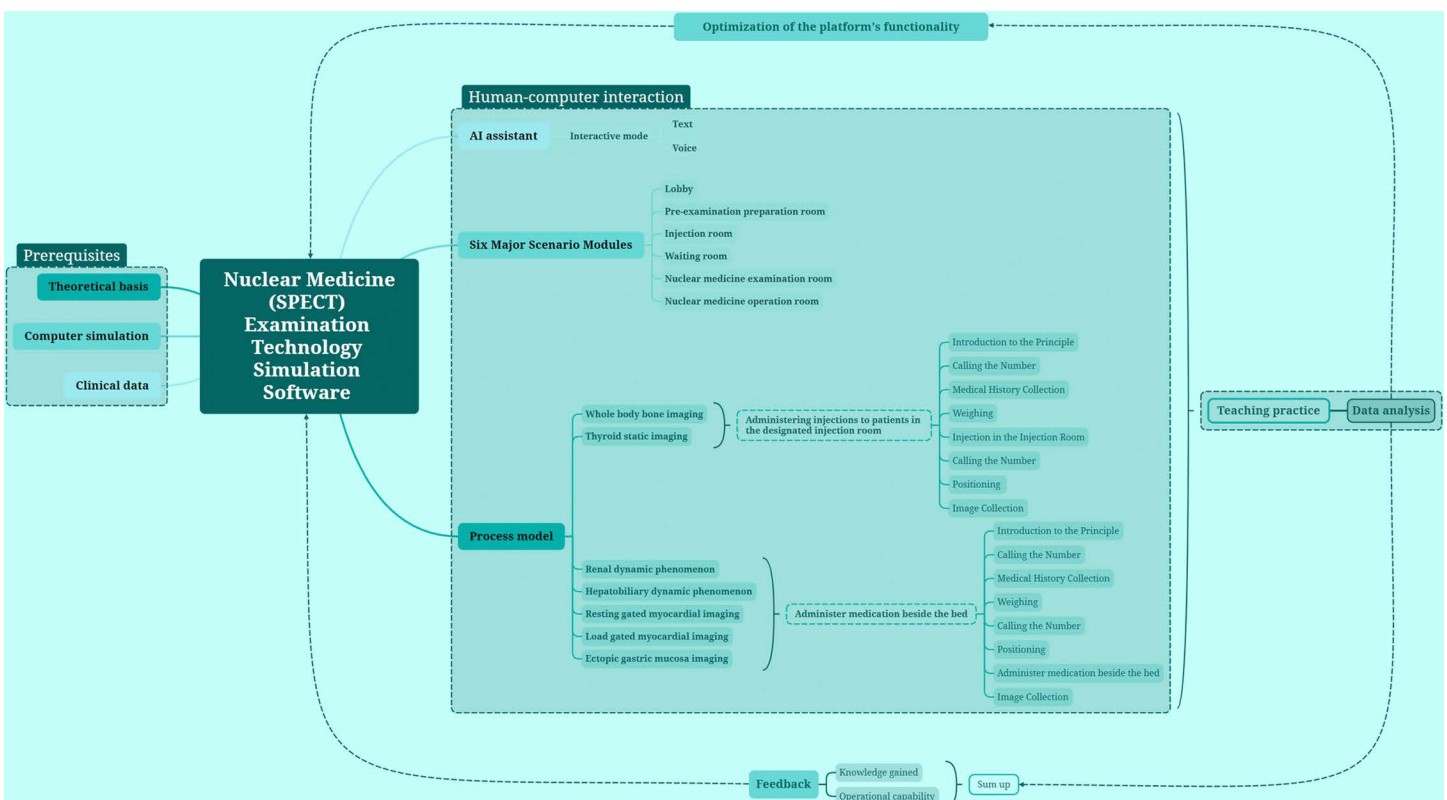

**Fig 2. Model design concept diagram.** Prerequisites, human-computer interaction, and feedback optimization.

The system's development technology employs Unity3D to construct 3D scenes and formulate interactive logic, which are then displayed and interacted with via a web browser using HTML5 technology. The AI assistant function was created utilizing the Python language and the Kouzi platform. Visual Studio, serving as the development environment, offers robust code editing and debugging capabilities. Concurrently, the Access database is employed for server-side data storage and management. The successful development and deployment of the online experimental simulation system were achieved through the synergistic use of these tools and technologies. Technical details are provided in Table 1.

**Model structure.**

(1) AI Assistant Module

The study's model system incorporates an AI assistant module, depicted in Fig 3. Students have the opportunity to engage with this AI assistant in two modalities: text and voice. Throughout the experimental procedure, students can utilize the AI assistant to address any queries they may have, thereby accommodating their individual learning requirements.

(2) Six Major Scenario Modules

The system comprises six primary scenario modules: the lobby, pre-examination preparation room, injection room, waiting room, scanning room of SPECT, and Operation room of SPECT. These modules are designed to offer students a comprehensive understanding of various scenarios and examination processes, thereby providing them with foundational

**Table 1. Details of the developed techniques.**

| Technology | Tools | Quality | Image Refresh Time | Pixel | Language | Database |
|---|---|---|---|---|---|---|
| 3D simulation, animation technology, WebGL technology, large model technology | Unity, Visual Studio, Buckle platform | 1080p | 60fps | 1024×960 | C#, python | Access, Mysql |

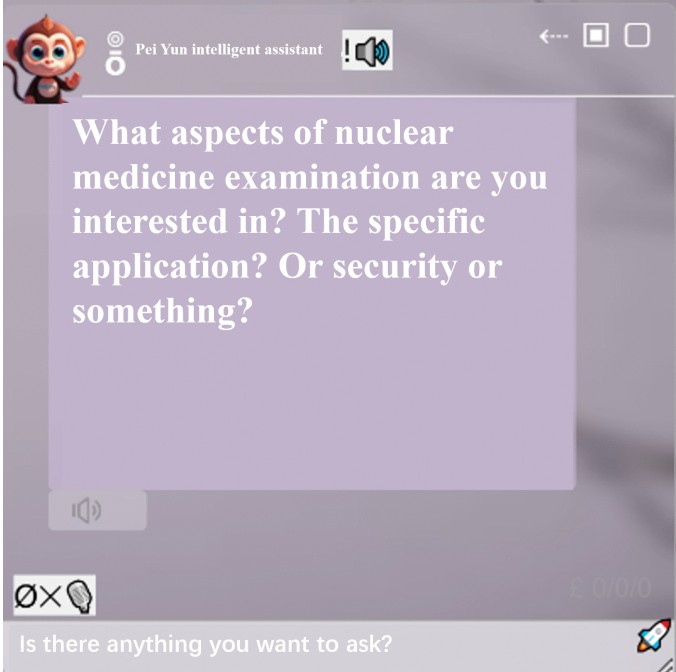

**Fig 3. AI assistant.** The AI assistant is represented by an icon of a small monkey.

knowledge of the environment and procedures for subsequent clinical practice. This approach addresses the challenge of gaining only observational experience with limited hands-on practice. Please refer to Fig 4 for further details.

① The hall features a triage desk, waiting chairs, landmarks, and simulated individuals, among other elements. As illustrated in Fig 5. ② The pre-examination preparation room is outfitted with a consultation desk, a consultation chair, a number calling button, a dialogue button, an inspection form, and a weighing scale, among other equipment. As illustrated in Fig 6. ③ The injection room is equipped with a lead protective cover, radioactive drugs, needles, and lead recycling buckets, as illustrated in Fig 7. ④ The waiting room is equipped with chairs for patients and features landmarks, as illustrated in Fig 8. ⑤ The scanning room of SPECT is equipped with both examination and supporting apparatuses. ⑥ The operation room of SPECT is equipped with a technician's console, a call button, a computer monitor, a keyboard, among other tools. This can be illustrated as depicted in Fig 9.

(3) Process model

The system encompasses seven distinct disease examination procedures,: thyroid static imaging, gated static myocardial tomography/stress myocardial tomography (also known as dynamic myocardial imaging), ectopic gastric mucosa imaging, hepatobiliary dynamic imaging, renal dynamic phenomenon, and whole-body bone imaging. The nuances in these examination differences are elucidated in Table 2.

The examination procedure comprises eight stages: Introduction to the Principle--Calling the Number--Medical History Collection--Weighing--Injection in the Injection Room (required for static collection)--Calling the Number--Positioning (for dynamic collection, an additional bedside medication step is needed)--Image Collection. An operational design of this examination process is illustrated in Fig 10.

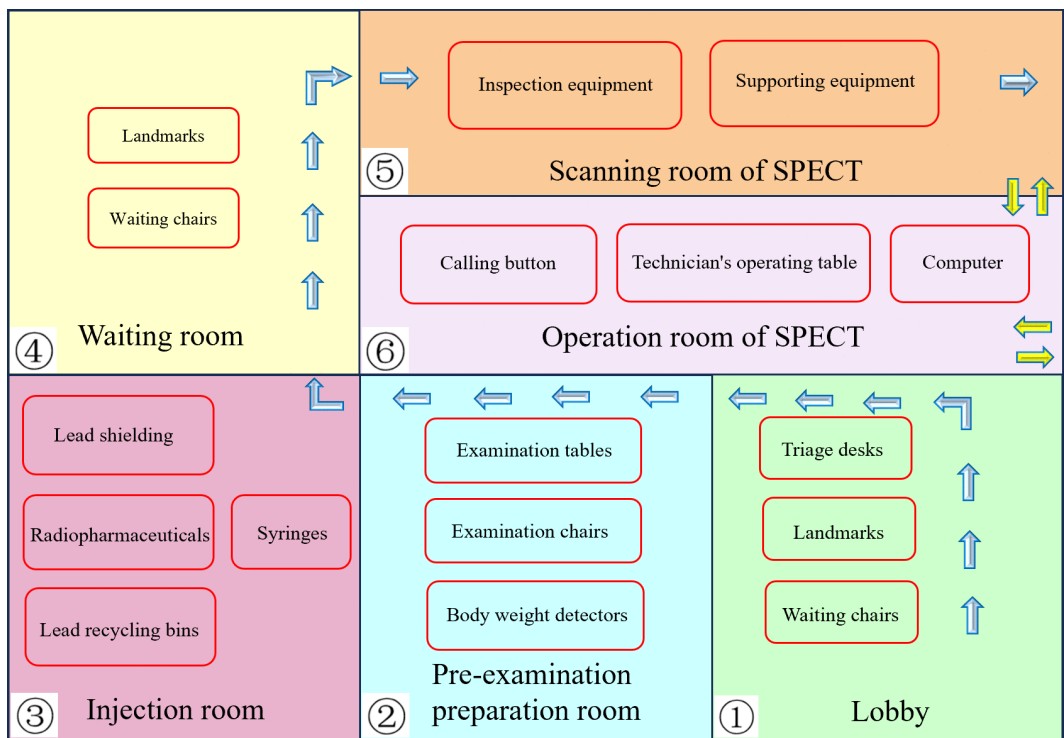

**Fig 4. Six major scenario modules.** Each box represents a scene module. Blue arrows depict the patient's route, while yellow arrows denote the nuclear medicine technologist's pathway.

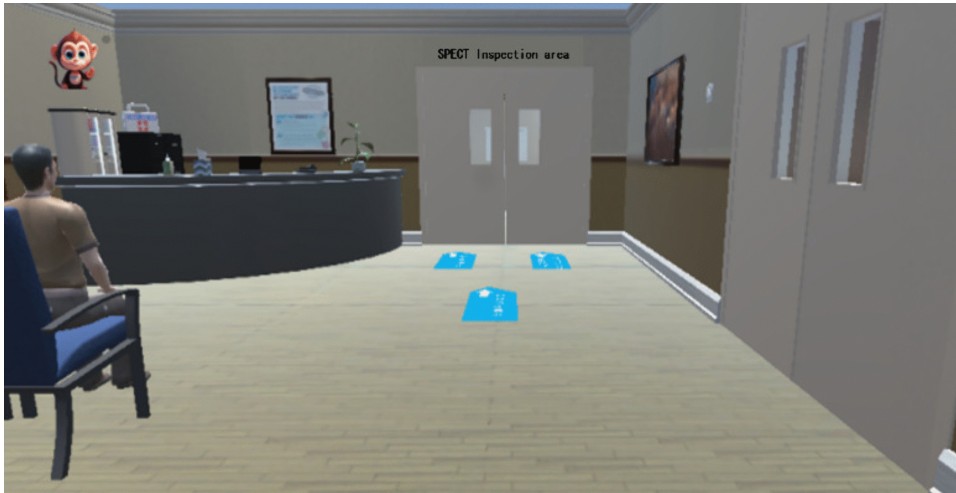

**Fig 5. Virtual interface of the hall.** The patient remains in the waiting hall, which is equipped with waiting desks and landmarks.

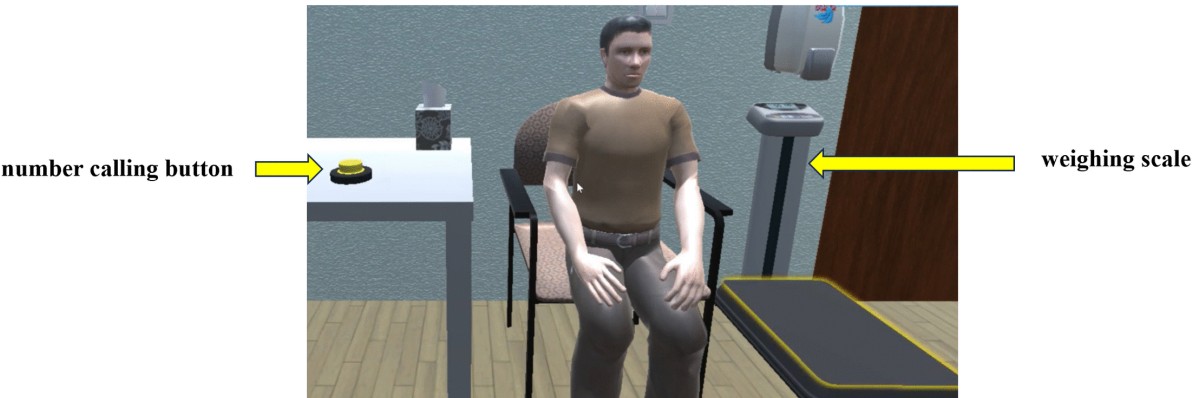

**Fig 6. The pre-examination preparation room.** In the image, a yellow call button is located on the left side of the consultation desk, while a weighing scale is situated on the right side.

The virtual experiment described herein requires a laboratory equipped with internet access and computers. The procedure is divided into eight steps, wherein students assume the dual roles of nuclear medicine technologists and patients, navigating the SPECT examination process through human-computer interaction. The theoretical introduction section provides explanations and exercises focused on the imaging principle of SPECT, thereby deepening the students' comprehension through a combination of learning and practice testing. The 'call out' and medical history collection sections are dedicated to the selection of examination forms and the completion of pre-examination precautions. The weighing section finalizes the allocation of the radiopharmaceutical injection dose. For static acquisition examinations, the radiopharmaceutical is administered in a separate injection room; the patient then waits in a designated waiting area until called out for the examination. In contrast, dynamic acquisition examinations proceed directly to the SPECT examination room after patient call out, where they recline on an examination bed and receive medication. Meanwhile, Nuclear medicine technologists conduct image acquisition on patients by remotely operating the equipment from within a shielded compartment. Finally, it is the students' responsibility to diagnose the collected images as either normal or abnormal and, through human-computer dialogue, inform patients of post-examination precautions. This concludes the experiment.

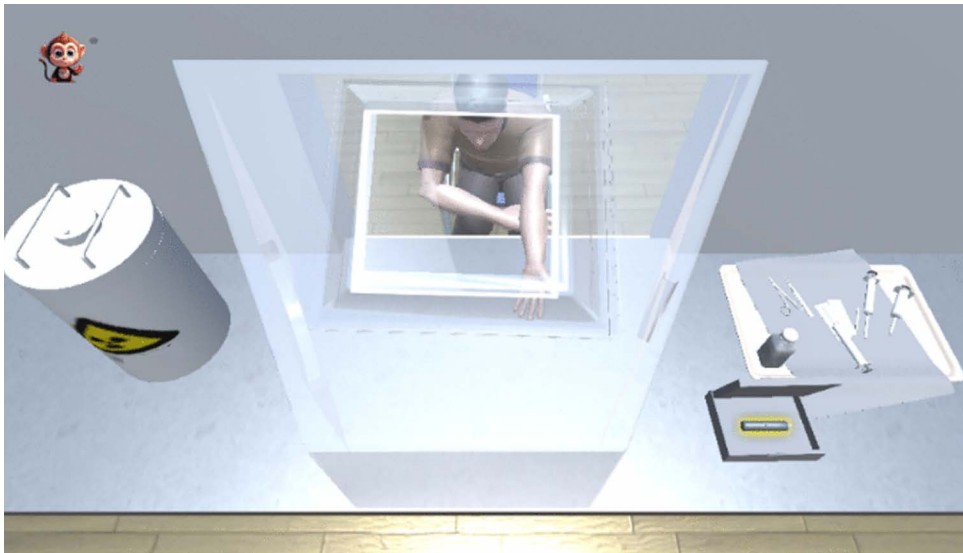

**Fig 7. The injection room.** The lead box, containing the radioactive drug, is situated on one side, while the patient waits on the opposite side of the lead protective cover for injection with the medication.

**(A)**  **(B)**

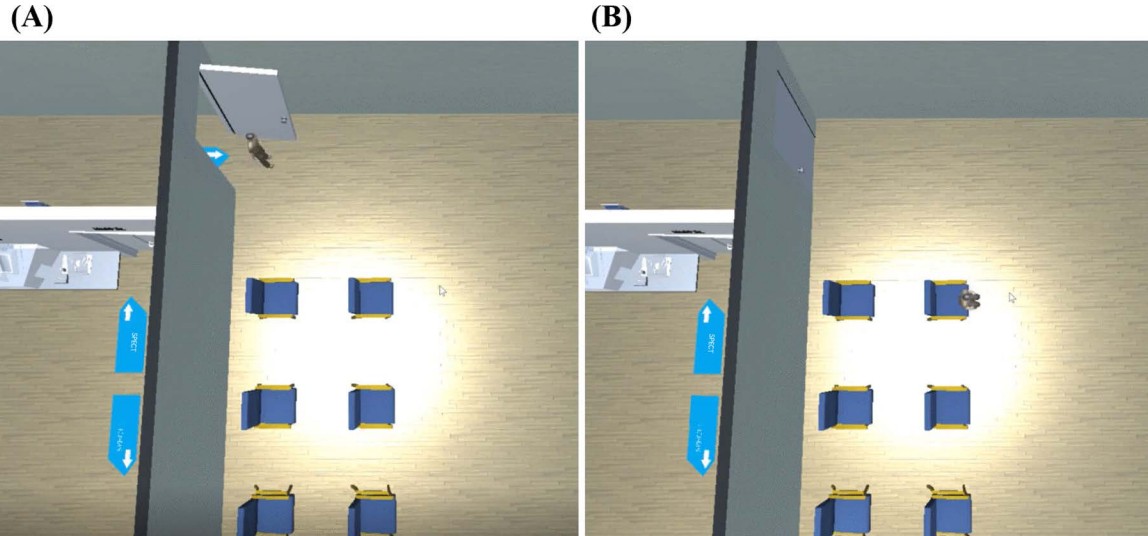

**Fig 8. The waiting room. (A)** The patient is entering the waiting room. **(B)** The patient is sitting in the waiting chair and waiting to be called.

The process model's characteristics primarily encompass five facets. ① The model comprises a training and an assessment mode. The training mode accommodates multiple tests to cater to students with diverse learning paces, enabling repeated practice. In contrast, the assessment mode is a one-time event, designed to encourage students to engage in frequent practice to attain their personalized learning objectives. ② The platform incorporates a role-playing feature, wherein students assume roles such as nuclear medicine technicians and patients, with the ability to switch between roles. ③ The system features a "level-up mode", demanding accurate operations for progression, thereby showcasing the system's hierarchical structure. ④ In instances of operator error, the "operation hint box" offers corrective suggestions,

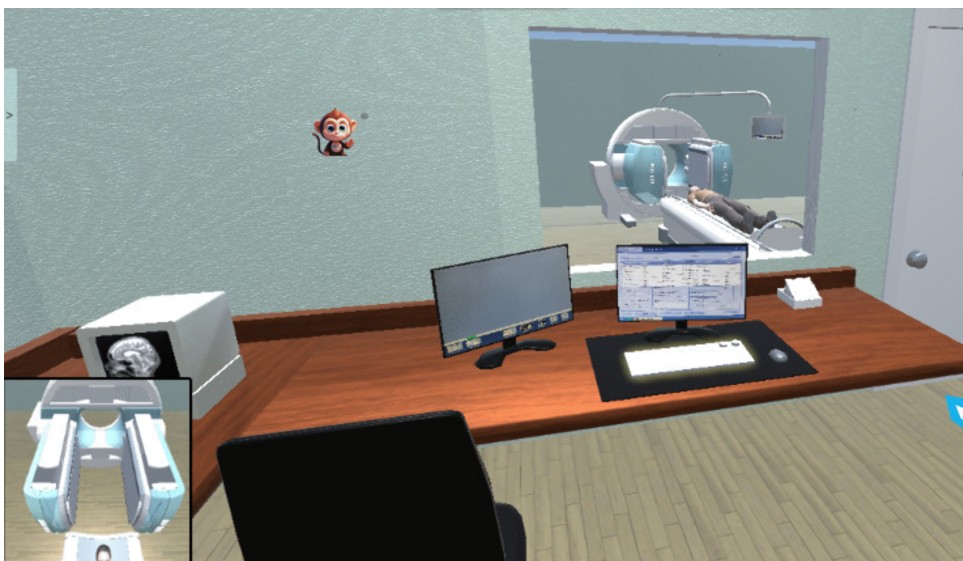

**Fig 9. The operation room.** Nuclear medicine technologists conduct image acquisition on patients by remotely operating the equipment from within a shielded compartment.

Table 2. The seven major inspection details.

| Check name | Injected drug | Injected dose (mCi) | Mode of injection | Empty stomach |
|---|---|---|---|---|
| **Whole body bone imaging** | $^{99m}$Tc-MDP | 15-25 | Administering injections to patients in the designated injection room. | No |
| **Thyroid static imaging** | $^{99m}$TcO$_4^-$ | 2--5 | Administering injections to patients in the designated injection room. | No |
| **Renal dynamic phenomenon** | $^{99m}$Tc-DTPA | 3--10 | Administer medication beside the bed | No |
| **Hepatobiliary dynamic phenomenon** | $^{99m}$Tc-PMT or $^{99m}$Tc-EHIDA | 5--10 | Administer medication beside the bed | Yes |
| **Resting gated myocardial imaging** | $^{99m}$Tc-MIBI | 20--25 | Administer medication beside the bed | Yes |
| **Load gated myocardial imaging** | $^{99m}$Tc-MIBI | 20-25 | Administer medication beside the bed | Yes |
| **Ectopic gastric mucosa imaging** | $^{99m}$TcO$_4^-$ | 10 | Administer medication beside the bed | Yes |

catering to the self-learning preferences of students. ⑤ Lastly, the system supports dynamic evaluation throughout the process, permitting educators to monitor and provide real-time feedback on students' learning progress.

## Teaching practice

The simulation system has been employed for instructional purposes across five academic cycles at Hebei Medical University, benefiting a total of 550 students. The participants were second-year university students, with an average age range of 18–20 years and a gender ratio of 57.1% (36,3% male and 63,6% female). Given that this SPECT virtual experiment constituted the inaugural instance at the institutional level, the participating students lacked prior exposure to such virtual simulations involving SPECT methodologies. Detailed statistics are provided in Table 3. Practice data for all students is provided in S1 Text.

In the second semester of 2024–2025, we established an intelligent course named "Principles of Medical Imaging" and integrated it with SPECT virtual simulation experiments to construct a ternary classroom. The ternary classroom approach

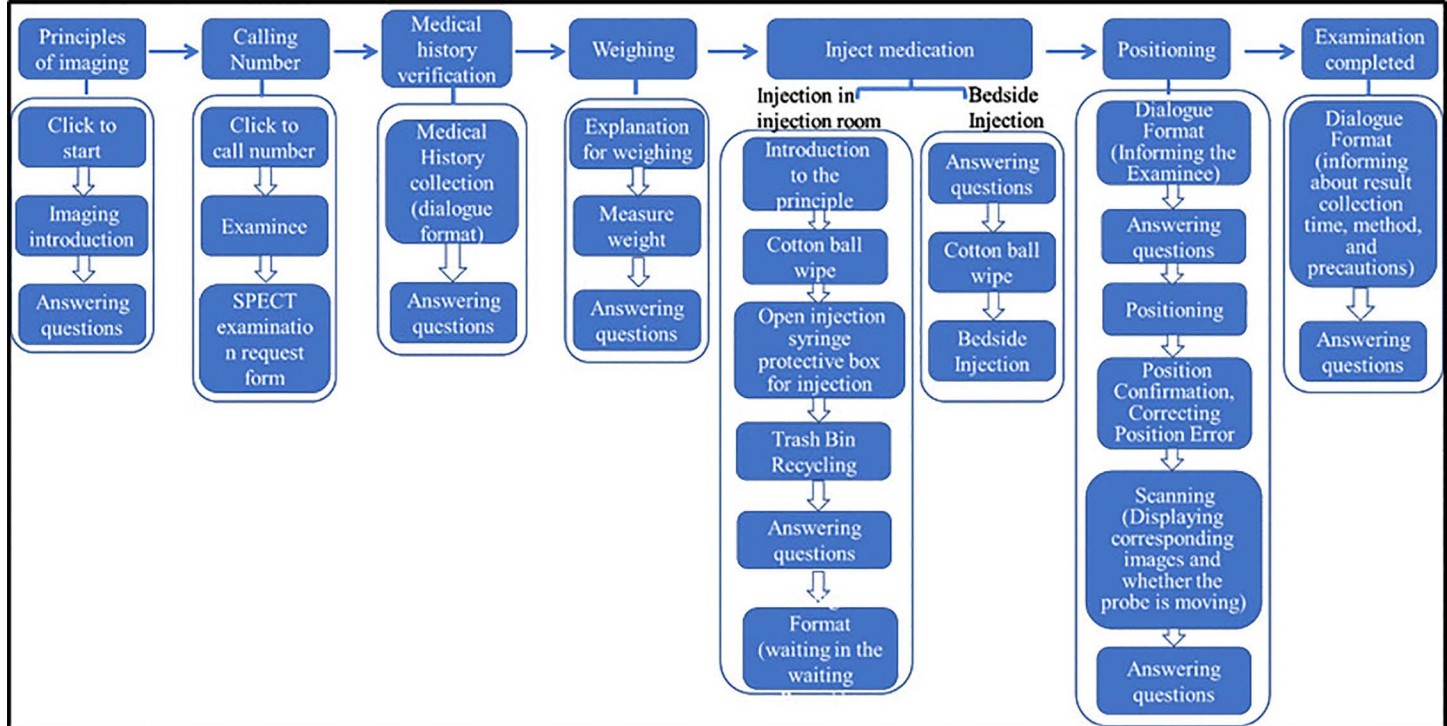

**Fig 10. SPECT operation flowchart.** The examination procedure comprises eight stages: Introduction to the Principle--Calling the Number--Medical History Collection--Weighing--Injection in the Injection Room (required for static collection)--Calling the Number--Positioning (for dynamic collection, an additional bedside medication step is needed)--Image Collection.

**Table 3. Details of teaching practice.**

| Time | Courses | Classes | Labora-tory hours | number of students |
|------|---------|---------|-------------------|--------------------|
| 2021-2022-1 | Nuclear medicine equipment and examination techniques | Grade 2019 Medical Imaging Technology Major | 3 | 102 |
| 2021-2022-2 | Principles of medical imaging | Grade 2020 Medical Imaging Technology Major | 3 | 146 |
| 2023-2024-1 | Principles of medical imaging | Grade 2022 Smart Medical Engineering | 3 | 30 |
| 2023-2024-2 | Principles of medical imaging | Grade 2022 Medical Imaging Technology Major | 3 | 143 |
| 2024-2025-2 | Principles of medical imaging | Grade 2023 Medical Imaging Technology Major | 3 | 129 |

encompasses pre-class guidance, in-class mutual learning, and after-class research. By offering teaching resources in a phased and tiered manner, this methodology caters to the individualized training requirements of students.

"Pre-class guidance" refers to the practice of providing students with the opportunity to conduct preliminary previews of the course material. This can be achieved through a variety of both online and offline resources, including knowledge maps, artificial intelligence (AI) assistants, and other digital materials.

The term "interactive learning during the course" pertains to the utilization of the SPECT virtual simulation experimental platform and interactive comments within nuclear medicine technologist classrooms. This approach facilitates a comprehensive range of learning activities, from theoretical study and equipment overview to examination procedures and image diagnosis. It employs a phased teaching methodology that enables students to systematically organize knowledge points and draw connections between problems. In essence, the entire classroom teaching process is structured in a phased,

layered, and goal-oriented manner. This methodology effectively enhances students' knowledge, abilities, and overall academic performance.

After-school research learning primarily pertains to students' utilization of reference books, academic literature, AI assistants, and digital resources post-class, enabling them to delve deeper into their studies and reach advanced levels of understanding.

## Results

### Comparative analysis of pre-class and post-class test scores

In contrast to the conventional teaching model, the pre-class test administered prior to the implementation of the SPECT inspection technology online training platform serves as a gauge of students' knowledge acquisition within the traditional pedagogical framework. Conversely, the post-class test, administered following the integration of this online platform, evaluates students' understanding within the innovative teaching paradigm of this study. Specifically, data from the 2023 cohort of medical imaging technology students was analyzed, as depicted in Fig 11. The mean score of the pre-class test was 85.98 (Fig 11.A), while the post-class test yielded an average score of 94.92 (Fig 11.B). This comparison reveals a 8.94% increase in knowledge acquisition by students utilizing the online training platform compared to their counterparts who did not. Furthermore, the post-class test of utilizing the online training exhibits a larger median and a smaller variance compared to the pre-class test. Please refer to Table 4 for more details.

### Evaluation of overall sample performance distribution

The statistical analysis presented in Table 5 is derived from the teaching practice results of the 2023-2024-2 and 2024-2025-2 semesters. It evaluates the performance of 272 samples, which were subjected to both the training mode (including the first training) and the assessment mode. The findings are subsequently elaborated upon in the ensuing paragraph.

The distribution of initial training scores for the entire sample predominantly aligns with the normal distribution, as depicted in Fig 12.A. This suggests that the question formulation on the platform is relatively appropriate.

(A)                                                                 (B)

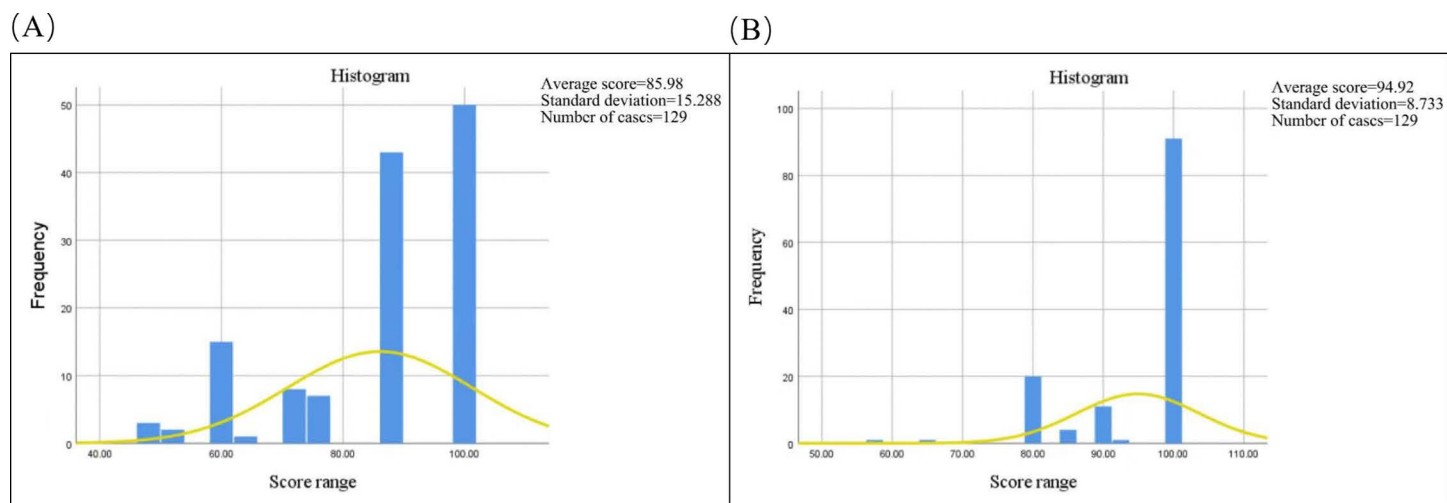

**Fig 11. Pre-class and post-class test.** The pre-class test and post-class test quantify the students' performance before and after utilizing the online practical training platform, respectively. A: Pre-class test scores; B: Post-class test scores.

**PLOS One**

**Table 4. Details of pre-class test and post-class test.**

| Grade category | | Pre-class test | Post-class test |
|---|---|---|---|
| Number of Students | | 129 | 129 |
| Average value | | 85.9845 | 94.9225 |
| Median | | 88.0000 | 100.0000 |
| Mode | | 100.00 | 100.00 |
| Standard Deviation | | 15.288 | 8.733 |
| Variance | | 233.719 | 76.263 |
| Skewness | | −0.946 | −1.656 |
| Standard Error of Skewness | | 0.213 | 0.213 |
| Kurtosis | | −0.211 | 2.363 |
| Standard Error of Kurtosis | | 0.423 | 0.423 |
| Range | | 52.00 | 42.50 |
| Total Sum | | 11092.00 | 12245.00 |
| Percentile | 10 | 60.0000 | 80.0000 |
| | 20 | 72.0000 | 85.0000 |
| | 30 | 88.0000 | 100.0000 |
| | 40 | 88.0000 | 100.0000 |
| | 50 | 88.0000 | 100.0000 |
| | 60 | 88.0000 | 100.0000 |
| | 70 | 100.0000 | 100.0000 |
| | 80 | 100.0000 | 100.0000 |
| | 90 | 100.0000 | 100.0000 |

The mean score of the initial training session was 85.12 points, exhibiting a standard deviation of 15.137, as evidenced by Fig 12.A. Subsequent to numerous training sessions, the mean score rose to 91.39 points, with a reduced standard deviation of 7.997, as evidenced by Fig 12.B, suggesting that repeated training significantly enhances student performance. Following these multiple training sessions, the mean sample score for the assessment was noted at 95.24 points, showcasing a standard deviation of 8.562, as evidenced by Fig 12.C. The distribution of assessment scores post-training demonstrated an exponential increase. This resulted in a more concentrated distribution of assessment scores in the high-score area, as evidenced by Fig 12.C.

### Evaluation of individual sample performance outcomes

Three students' scores were randomly selected as research samples, and the distribution of their multiple training scores was analyzed. The results are presented in Fig 13.

The statistical analysis reveals a significant positive correlation between the increase in training session numbers and sample scores. This suggests that the training model substantially enhances performance metrics, thereby affirming its value.

In Sample 1 (Fig 13.A), corresponding to Student ID 22011220011, the subject underwent seven training sessions, yielding an average score of 98.86. The slope of the linear fit of these scores is 0.7500.

Sample 2 (Fig 13.B) pertains to a student identified as ID 22011220013, who underwent eight training sessions, yielding an average score of 96.88. The slope of the linear fit of these scores is 1.35.

Sample 3 (Fig 13.C) pertains to a student identified by ID 22021070017, who underwent eight training sessions, yielding an average score of 94.63. The linear fit slope of the scores is notably 2.71.

**Table 5. Results of training and assessment mode.**

| Grade category | | First training result | Multiple training average | Assessment results |
|---|---|---|---|---|
| Number of Students | | 270 (2 missing) | 270 (2 missing) | 272 |
| Average value | | 85.1222 | 91.3922 | 95.2390 |
| Median | | 89.5000 | 93.8333 | 100.0000 |
| Mode | | 100.00 | 92.00 | 100.00 |
| Standard Deviation | | 15.137 | 7.997 | 8.562 |
| Variance | | 229.126 | 63.947 | 73.304 |
| Skewness | | −2.236 | −2.353 | −3.02 |
| Standard Error of Skewness | | 0.148 | 0.149 | 0.148 |
| Kurtosis | | 8.491 | 6.382 | 12.132 |
| Standard Error of Kurtosis | | 0.295 | 0.296 | 0.294 |
| Range | | 100.00 | 49.00 | 65.00 |
| Total Sum | | 22983.00 | 24584.51 | 25905.00 |
| Percentile | 10 | 66.0000 | 81.8000 | 88.0000 |
| | 20 | 74.0000 | 88.2000 | 91.0000 |
| | 30 | 82.0000 | 91.2857 | 96.0000 |
| | 40 | 85.4000 | 92.6364 | 98.0000 |
| | 50 | 89.5000 | 93.8333 | 100.0000 |
| | 60 | 93.0000 | 95.0000 | 100.0000 |
| | 70 | 93.0000 | 95.7143 | 100.0000 |
| | 80 | 97.0000 | 96.6250 | 100.0000 |
| | 90 | 100.0000 | 97.5714 | 100.0000 |

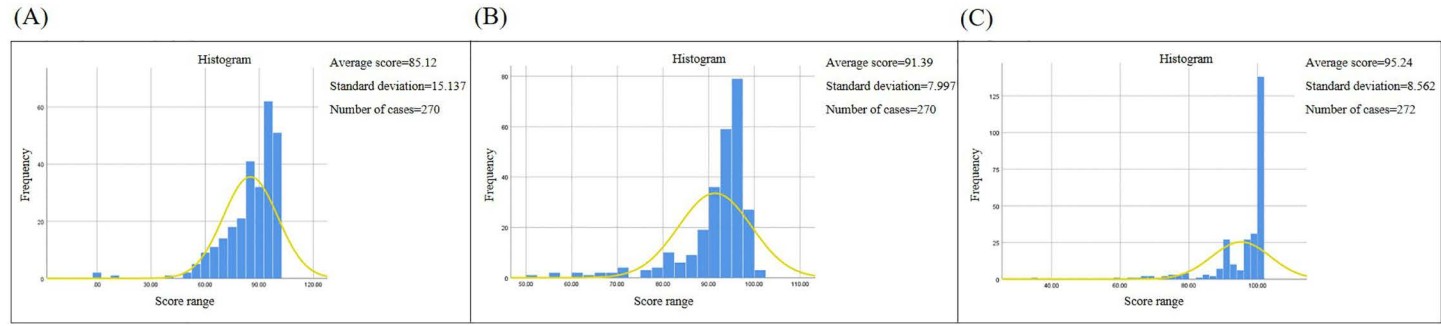

**Fig 12. Overall sample performance distribution.** The dispersion of student grades across 272 total samples for the 2023-2024-2 and 2024-2025-2 academic year's second semesters is detailed.A: First training result.B:Multiple training average.C:Assessment results.

## The impact of personalized training times on student assessment performance

The data presented in Fig 14 demonstrates a marked improvement in assessment scores compared to pre-training values. It is evident that, after five sessions of training, scores can attain a high level.

## Course satisfaction survey

Course satisfaction was evaluated using a survey question administered through Wjx: "Are you satisfied with the teaching methods used in the 'Principles of Medical Imaging' course?" Statistical analysis revealed that 94.64% of students expressed satisfaction with the pedagogical approaches employed,as illustrated in Fig 15.

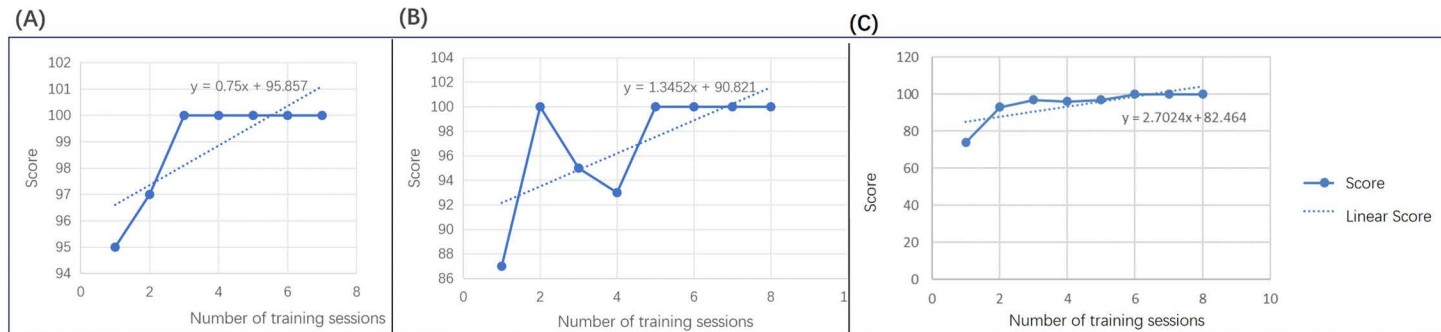

**Fig 13. Individual sample grade distribution.** Three distinct samples were arbitrarily chosen for examination. The horizontal axis indicates the frequency of training sessions, while the vertical axis denotes performance. A: Sample 1.B:Sample 2.C:Sample 3.

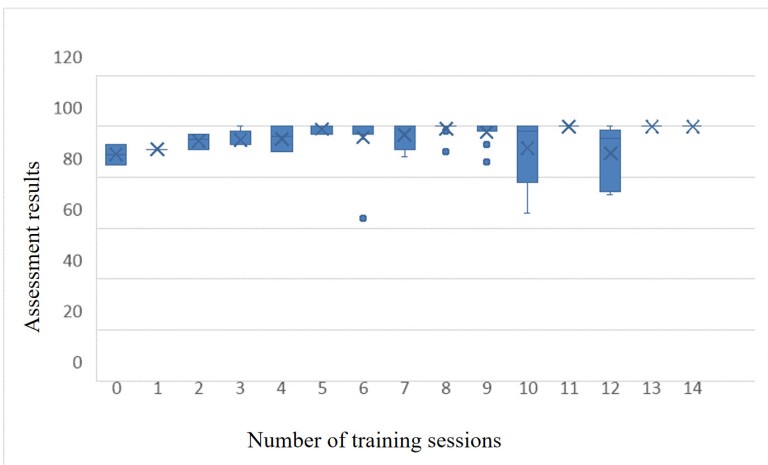

**Fig 14. The effect of training frequency.** The horizontal axis represents the number of training sessions, while the vertical axis represents the assessment results.

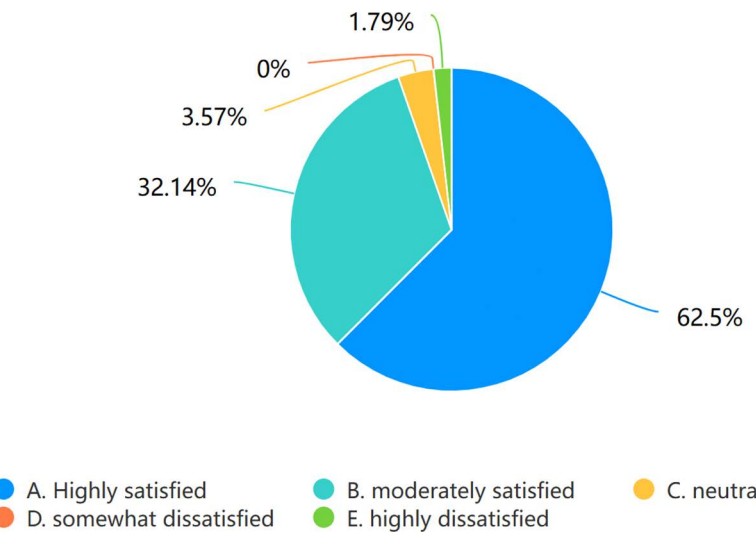

**Fig 15. Course satisfaction survey.** Different colors denote varying degrees of satisfaction.

## Discussion

### Comparative analysis of pre-class and post-class test scores

An analysis of pre-class test scores (traditional mode) and post-class test scores (reformed mode) indicates that the online training platform enhances students' learning efficacy, thereby achieving the anticipated research objectives.

### Evaluation of overall sample performance distribution

The comprehensive statistical analysis reveals that the score distribution from the initial training session (Fig 12.A), along with the average scores across multiple training sessions (Fig 12.B), align closely with a normal distribution. This suggests that the quantity and difficulty of questions posed by the platform are relatively balanced and appropriate.

After multiple training sessions, the average score of students increased by 6.27, indicating that repeated training is beneficial for the improvement of student performance. Post-training, there was a marked exponential increase in the distribution of test scores, suggesting that repeated training rounds significantly enhance test performance. Furthermore, there was a heightened concentration of test scores in the upper range, as depicted in Fig 12.C.

The average score of the initial training session was 85.12 points. This can be attributed to the fact that foundational procedural tasks, such as selecting an injection method and removing foreign objects, are relatively simple assessments and thus do not significantly differentiate participants. Furthermore, these basic functional capabilities do not necessitate prolonged training. However, more advanced capabilities cannot be met without appropriate training.

The differentiation is primarily evident in tasks such as bedside injection operations and scan positioning, which necessitate a high level of knowledge mastery. These tasks not only reflect the proficiency in inspection techniques but also enhance it through training. This component is the heart of inspection technology, suggesting that skills can only be honed through meticulous practice and instruction. This underscores the value of this assessment system, as it effectively differentiates core skill proficiency.

### Evaluation of individual sample performance outcomes

Upon analyzing the results of individual samples, it was discovered that, the statistical analysis reveals a significant positive correlation between the increase in training session numbers and sample scores. This suggests that the training model substantially enhances performance metrics, thereby affirming its value.The lesser slope exhibited by Sample 1 (Fig 13.A) can be attributed to the robust foundation and strong practical skills of this sample, which yielded high yet somewhat variable scores. Nevertheless, the positive slope of the score indicates that the training regime also contributes to maintaining score stability. The score curve for Sample 2 (Fig 13.B) shows more pronounced fluctuations. This is primarily due to the system's training mode, which employs a random extraction method; thus, the content and operational procedures differ with each inspection. Consequently, the same learner might encounter variations in their scores. However, as they become more familiar with diverse inspections, the score distribution evens out. Sample 3 (Fig 13.C) initially displayed lower scores, but demonstrated considerable improvement after fewer training sessions, resulting in a steeper score slope. This suggests rapid mastery of the learning or training techniques.

### The impact of personalized training times on student assessment performance

The personalized training times have a positive feedback effect on student assessment scores. This indicates that the design of training and assessment modes is reasonable and can meet the individual learning needs of students.

### Course satisfaction survey

The data of 94.64% satisfaction strongly suggests that the teaching model is highly regarded by students. Consequently, this pioneering model accommodates the requisites of student progression and pedagogical transformation.

## Conclusion

Compared to traditional theoretical instruction, the online training platform for SPECT examination technology developed by our team enhances practical teaching of SPECT within the academic setting. This virtual practice is radiation-free and not constrained by time or space, allowing for an integrated theory-practice pedagogical approach. Data from pre- and post-class assessments demonstrate that this method has significantly improved students' understanding. Furthermore, the model leverages virtual simulation technologies to restructure educational content, facilitating a comprehensive review of nuclear medicine technology. It caters to individualized training needs, bolsters the systematic nature of students' knowledge acquisition, and refines the educational framework. By incorporating modern digital tools like AI assistants and knowledge graphs, we have achieved a profound integration of digital technology and educational instruction.

The platform's application has thus far been confined to the domain of PET imaging (Further details are provided in S2 Text). However, over the course of the next five years, its utilization is anticipated to expand to SPECT/CT imaging. Further advancements are expected in the subsequent decade, with potential integration into the SPECT/MR imaging field. The platform, currently extended to three additional medical colleges (Further details are provided in S2 Text), has garnered significant commendation from peer institutions. Looking ahead, it is projected that within five years, the platform will facilitate the training of undergraduate students across ten medical colleges and residents in affiliated hospitals. Within a decade, this figure is expected to rise to encompass 50 medical colleges or affiliated hospitals. Notably, continual iterative updates will be made to the platform to bolster its functionality. For instance, there will be an increase in the number of cases to strengthen students' early clinical job competencies [28–34]. Additionally, the platform aims to optimize the natural interaction between patients and technicians, thereby fostering students' communication skills, emotional intelligence, and psychological development [35–40]. Furthermore, a voice guidance function will be incorporated to assist students with reading disabilities, such as those with poor eyesight, in navigating the platform [41]. In conclusion, teaching innovation represents not just the convergence of theory and practice, technology and science, but also the forefront of medical advancement and human health protection. In our teaching reform, we steadfastly adhere to the principle of "having levels, having connotation, having goals", with the ultimate objective of nurturing students into a new generation of technically proficient medical imaging professionals who "know the principles, understand technology, and can create".

## Supporting information

**S1 Text. Dataset.** Practice data for all students.
(ZIP)

**S2 Text. Supporting information.**
(PDF)

## Acknowledgments

We would like to express our gratitude to the Second Hospital of Hebei Medical University for providing us with valuable clinical data. Furthermore, we acknowledge the technical support provided by Shanghai Peiyun Educational Technology Co., Ltd.

## Author contributions

**Conceptualization:** Hongzhi Wang, Ruibin Zhao, Dandan Shang.

**Data curation:** Lihua Qiao, Hongzhi Wang, Xinkun Lei, Tian Xia, Zikun Fang, Luqi Shou, Yiwen Qin.

**Formal analysis:** Lihua Qiao, Xiaorui Guo, Tian Xia, Yiwen Qin.

**Funding acquisition:** Lihua Qiao, Hua Shang, Tian Xia, Dandan Shang.

**Investigation:** Hongzhi Wang, Hua Shang, Tian Xia.

**Methodology:** Lihua Qiao, Hua Shang, Tian Xia.

**Project administration:** Lihua Qiao, Xinkun Lei, Hua Shang, Tian Xia, Luqi Shou, Dandan Shang.

**Resources:** Hongzhi Wang, Ruiping Qin, Dandan Shang.

**Software:** Xiaorui Guo, Ruiping Qin.

**Supervision:** Xiaorui Guo, Xinkun Lei, Ruibin Zhao, Ruiping Qin, Zikun Fang, Luqi Shou.

**Validation:** Hongzhi Wang, Xiaorui Guo, Hua Shang, Ruibin Zhao, Ruiping Qin, Zikun Fang, Luqi Shou, Yiwen Qin.

**Visualization:** Xinkun Lei, Ruibin Zhao, Zikun Fang, Yiwen Qin, Dandan Shang.

**Writing – original draft:** Lihua Qiao, Dandan Shang.

**Writing – review & editing:** Lihua Qiao, Ruibin Zhao, Dandan Shang.

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
