## [Decision Letter · Decision Letter 0]

20 May 2025

Dear Dr. Shang,

Thank you for submitting your manuscript to PLOS ONE. After careful consideration, we feel that it has merit but does not fully meet PLOS ONE’s publication criteria as it currently stands. Therefore, we invite you to submit a revised version of the manuscript that addresses the points raised during the review process.

Please address the reviewers’ comments and resubmit within the given timeframe. As optional enhancements, you may include an Author Contributions statement, streamline your reference list to the most pertinent sources, and adjust paragraph structure to improve readability. I look forward to receiving your revised manuscript.

We look forward to receiving your revised manuscript.

Kind regards,

Alexandre Bonatto

Academic Editor

PLOS ONE

Journal Requirements:

Research and Practice on Innovation and Entrepreneurship Education and Teaching Reform in Universities of Hebei Province in 2023 (2023cxcy068),, Medical Science Research Project of Hebei Province in 2022 (20220971), Research and Practice on Higher Education Teaching Reform in Hebei Province (2022GJJG149), Educational and Teaching Research Project of Hebei Medical University in 2022 (2022YBZD-4, 2022YBPT-8), and University Student Innovation Experiment Plan Project of Hebei Medical University in 2023 (USIP2023337).

Additional Editor Comments :

Dear Author,

Thank you for submitting your manuscript. After careful review, I am pleased to offer acceptance pending minor revisions: please address the reviewers’ comments and resubmit within 45 days. As optional enhancements, you may include an Author Contributions statement, streamline your reference list to the most pertinent sources, and adjust paragraph structure to improve readability. I look forward to receiving your revised manuscript.

Reviewers' comments:

Reviewer's Responses to Questions

**Comments to the Author**

1. Is the manuscript technically sound, and do the data support the conclusions?

Reviewer #1: Yes

Reviewer #2: Yes

2. Has the statistical analysis been performed appropriately and rigorously?

Reviewer #1: I Don't Know

Reviewer #2: Yes

3. Have the authors made all data underlying the findings in their manuscript fully available?

Reviewer #1: No

Reviewer #2: Yes

4. Is the manuscript presented in an intelligible fashion and written in standard English?

Reviewer #1: No

Reviewer #2: Yes

Reviewer #1: Hello, respected authors, and thank you for this interesting article. Although for various reasons I was against rejection, I think the authors can be given another chance.

A strange and thought-provoking point is the presence of more than ten authors for the article. Please justify that this number of authors is really necessary? And where did they contribute?

How interesting and interesting the topic is, but I offer a few points to improve your article that I hope will be useful and useful:

Reduce old references and do not exceed 20.

Do not write paragraphs that are too long or too short, try to maintain order.

Your statement of the problem is a little weak. Explain in one paragraph why this research is important and necessary.

Why did you choose Vietnam?

Your article is international. Can you achieve some generalization in the findings from a limited number of countries and individuals? Explain this a little bit about this issue as well.

Completely and several times to strengthen the structure of the journal and observe all its issues.

Explain whether the sample is sufficient for the research population and can it be strengthened?

In my opinion, strengthen the qualitative section of the article because it will provide readers with the existing attractions.

The limitations of the research are not included in the conclusion.

Add a section titled Discussion and compare all the research with previous research.

There needs to be some direct promise in the article, which has made the text very sweet, but it is more literary than scientific.

The sixth research did not study new research in this field and this makes your article look like an old article.

The article is too short, or you were in a hurry, or you could not spend the time as much as you would like.

Ten references are not enough for an original article, I think you should increase it to 40.

Please provide a timeline of the research process.

Best wishes

Reviewer #2: This is a well written manuscript that showcases the development and use of SPECT training, combining 3D modeling, AI assistance, and multimodal instructional strategies. The study presents both group-level and individual-level performance data, the “three-element classroom” (pre-class, in-class, and post-class) design is aligned with active learning best practices.

It would have been good if the authors could elaborate why a control group was not used or comparative data against traditional teaching methods was done.

While quantitative performance is analyzed, student perceptions, satisfaction, and engagement are only briefly mentioned. Please elaborate on these findings as well.

The manuscript would benefit from English editing to simplify the writing. Some sections (e.g., tables, explanations of AI functionality) are overly verbose or technically dense for a general readership.

What extent the platform could be generalized to other imaging modalities or clinical areas?

**Do you want your identity to be public for this peer review?** For information about this choice, including consent withdrawal, please see our Privacy Policy

Reviewer #1: **Yes:** Amir Karimi

Reviewer #2: No

---

## [Author Response · Author response to Decision Letter 1]

2 Jul 2025

Response to Reviewers

Dear Editor and Reviewers:

Thank you for your insightful feedback on this article. I will address the concerns of the article sequentially from the perspectives mentioned below.

1 Editor's queries

Question 1. Please ensure that your manuscript meets PLOS ONE's style requirements, including those for file naming.

Answer: The manuscript has been revised to conform to the style guidelines specified by "PLOS ONE". Please refer to the " Revised Manuscript with Track Changes" or " Manuscript -New " for further details.

Question 2. Please ensure that you have an ORCID iD and that it is validated in Editorial Manager.

Answer:The ORCID iD of the corresponding author has been duly added and verified in "Editorial Manager".

Question 3. Please state what role the funders took in the study. If the funders had no role, please state: "The funders had no role in study design, data collection and analysis, decision to publish, or preparation of the manuscript."If this statement is not correct you must amend it as needed.Please include this amended Role of Funder statement in your cover letter; we will change the online submission form on your behalf.

Answer:①The primary supporting project for this article is the "Research and Practice of Innovation and Entrepreneurship Education and Teaching Reform in Hebei Province's Universities in 2023 (2023CXCY068)". The project content aligns seamlessly with this article, focusing on the development and operation of a virtual experimental platform for nuclear medicine. This platform is designed to enhance the innovation and entrepreneurship training of professionals specializing in medical imaging technology. The findings of this article will be derived from the outcomes of the project.② The research concepts and program code for this study were supplied by the "Hebei Province Medical Science Research Project in 2022 (20220971)".③ This article's teaching innovation design and practice are founded on the project titled "Research and Practice of Higher Education Teaching Reform in Hebei Province (2022GJJG149)", a 2022 Education and Teaching Research Project from Hebei Medical University (2022YBZD-4, 2022YBPT-8). ④ Data analysis for this article is provided by the project "2023 College Students' Innovation Experiment Plan of Hebei Medical University (USIP2023337)".Consequently, all the aforementioned elements are pivotal to the content of this article and should be preserved.

Question 4. Please amend either the title on the online submission form (via Edit Submission) or the title in the manuscript so that they are identical.

Answer: Modified the title on the online submission form (by editing the submission) or in the manuscript, such that it is identical to the title in the revised manuscript.

Question 5. Please include captions for your Supporting Information files at the end of your manuscript, and update any in-text citations to match accordingly.

Answer: The supporting information file's title and content are provided at the conclusion of the manuscript, as outlined on page 34 of the " Revised Manuscript with Track Changes" within the "Supporting Information" section.

Question 6. Please review your reference list to ensure that it is complete and correct.。

Answer The references have been meticulously reviewed, and both the format and quantity of references have been appropriately revised in the "Revised Manuscript with Track Changes " on page 28, within the "References" section.

2 Questions Raised by Reviewer 1

Question 1. Please justify that this number of authors is really necessary? And where did they contribute?

Answer :The online training platform for SPECT imaging technology, developed by our research team, represents a collaborative effort among universities, hospitals, and private enterprises. University instructors Shang Dandan, Qiao Lihua, Zhao Ruibin, and Qin Ruiping brought innovative concepts, pragmatic solutions, course execution, and authored articles to the project. Hospital nuclear medicine technologists Guo Xiaorui, Lei Xinkun, and Shang Hua supplied clinical data, constructed scenes, and designed cases. Wang Hongzhi, a professor at East China Normal University and chairman of Shanghai Peiyun Educational Technology Co., Ltd., alongside enterprise engineer Xia Tian, offered technical assistance, including system debugging and maintenance. Current students Fang Zikun, Shou Luqi, and Qin Yiwen contributed to the analysis and feedback on teaching practice data. In essence, this research encompassed the efforts of various stakeholders, including multiple institutions, educators, students, and company staff, spanning from development to instructional implementation. Consequently, every author has played a pivotal role. Please refer to the“Author Contributions”section, which has been added on page 28 of the“Revised Manuscript with Track Changes”.

Question 2. Reduce old references and do not exceed 20.

Answer The number of outdated references has been suitably reduced to a maximum of 20, while the overall quantity of references has been augmented to 41, as indicated on page 28 "References" in the " Revised Manuscript with Track Changes ".

Question 3. Do not write paragraphs that are too long or too short, try to maintain order.

Answer Paragraphs of lesser length were amalgamated and excessively long sentences were judiciously excised. This can be observed in the "Six Major Scenario Modules" section on pages 9-11, the "Process model" section from pages 11-12, and in the " Revised Manuscript with Track Changes " where longer paragraphs have been condensed on pages 11 and 14.

Question 4. Your statement of the problem is a little weak. Explain in one paragraph why this research is important and necessary.

Answer The online training platform developed by our team for SPECT imaging technology addresses the high risks and costs associated with SPECT equipment, as well as the challenges of implementing experimental classes on campus. This platform offers a departure from traditional teaching constraints, allowing students to engage in educational practice without the limitations of time, space, or frequency, thereby accommodating their individualized needs. The platform's design is supported by training data which indicates that the overall sample score distribution aligns with a normal distribution. Furthermore, the positive slopes of individual sample scores suggest that the frequency of training sessions has a beneficial impact on students' personalized development. The integrated dynamic evaluation system provides constructive feedback on teaching outcomes, thereby achieving the objectives of nuclear medicine education in terms of personalized student cultivation, innovative thinking, and the development of an "early clinical" mindset.

Question 5. Why did you choose Vietnam?

Answer It appears to be an operational error; the study was conducted in the People's Republic of China instead of Vietnam as originally intended.

Question 6. Your article is international. Can you achieve some generalization in the findings from a limited number of countries and individuals? Explain this a little bit about this issue as well.

Answer An online training platform for Single-Proton Emission Computed Tomography (SPECT) examination technology, incorporating AI assistant functions and featuring seven typical disease examinations, could not be located in international data sets. This absence is consistent with the computer software copyright associated with this study. The comprehensive teaching innovation model, from the conceptualization and development of the online training platform to its implementation in pedagogical practice, has not been previously observed in related SPECT teaching methodologies. Thus, this study possesses significant originality.

Question 7. Completely and several times to strengthen the structure of the journal and observe all its issues.

Answer The paper has been restructured in accordance with your request and is now presented within the " Revised Manuscript with Track Changes ".

Question 8. Explain whether the sample is sufficient for the research population and can it be strengthened?

Answer To date, over 550 students from our institution have engaged in teaching practice on this platform, fully addressing the research objectives. This practice is executed in a phased manner, with class-specific instruction. Owing to space constraints within the article, data analysis was confined to students from a particular class in a specific semester. Furthermore, three additional universities are utilizing this platform. For further details, please see Attachment 2.

Question 9. In my opinion, strengthen the qualitative section of the article because it will provide readers with the existing attractions.

Answer The "Abstract" on page 3 and the "Discussion" on page 23 of the " Revised Manuscript with Track Changes " have been revised by incorporating qualitative speech.

Question 10. The limitations of the research are not included in the conclusion.

Answer The limitations have been excised from the conclusion section and relocated to the " Revised Manuscript with Track Changes " in the " Conclusion " section on page 25.

Question 11. Add a section titled Discussion and compare all the research with previous research.

Answer�A "Discussion" section has been incorporated on page 23 of the " Revised Manuscript with Track Changes ". Additionally, a comparative analysis with previous research has been included. The data relevant to this comparison is presented on page 17 and 23 of the " Revised Manuscript with Track Changes " within the Comparative Analysis of Pre-class and Post-class Test Scores.

Question 12. There needs to be some direct promise in the article, which has made the text very sweet, but it is more literary than scientific.

Answer In the " Revised Manuscript with Track Changes ", we have incorporated future planning commitments and application expansion commitments into the “Conclusion” section on page 25.

Question 13. The sixth research did not study new research in this field and this makes your article look like an old article.

Answer This study has secured a copyright for the computer software utilized in the research, as outlined in Appendix 1(S1 File), underscoring the novelty of the project. The development and implementation of an online training platform, specifically designed for educational purposes, represents the most comprehensive and systematic teaching system globally for conducting SPECT virtual experiments within an academic setting. Consequently, this study is both innovative and of significant practical value.

Question 14. The article is too short, or you were in a hurry, or you could not spend the time as much as you would like.

Answer The supplementary content is comprehensively outlined in the " Revised Manuscript with Track Changes ": on page 2 within the "Abstract" section; page 15, where "Table 3" now includes data for "2024-2025(2)"; page 17 and 23, which features an added section titled "Comparative Analysis of Pre-class and Post-class Test Scores"; page 18, showcasing updated "Table 4" data; and page 22 and 25, where three new subsections have been introduced: "(4) The influence of customized training sessions on student evaluation scores", "Course Satisfaction Survey", and page 25" Conclusion".

Question 15. Ten references are not enough for an original article, I think you should increase it to 40.

Answer The quantity of references has been augmented to a total of 42, as indicated within the "References" section on page 28 of the " Revised Manuscript with Track Changes ".

Question 16. Please provide a timeline of the research process.

Answer June 2020 - December 2020: The university, enterprise, and hospital collaborated to sign an agreement, determining the research plan, including platform design, data collection, virtual experimental scenario modules, and operational process module design, etc.

January 2021 - February 2022: The initial development of the SPECT examination technology online training platform was completed, and it was piloted during the 2021-2022 (1) semester.

March 2022 - July 2022: The SPECT inspection technology online training platform was officially developed and completed, and was approved for computer software copyright, officially being put into educational practice.

August 2022 - July 2023: Iterative optimization of the SPECT examination technology online training platform, organization and analysis of teaching practice data.

August 2023 - December 2023: A provincial first-class undergraduate virtual simulation experimental teaching course based on this platform was applied for and approved.

Jan 2024–Mar 2025: Writing of papers related to the study, completion and submission of articles..

3 Questions Raised by Reviewer 2

Question 1. It would have been good if the authors could elaborate why a control group was not used or comparative data against traditional teaching methods was done.

Answer In this study, a control group was not established, given the potential for individual students to attribute their performance outcomes either to the prohibition of using the training platform or its utilization. Instead, the research employed a comparative approach, examining the difference in knowledge mastery between pre-class and post-class assessments. Refer to page 17 and 23, "Comparative Analysis of Pre-class and Post-class Test Scores" in the " Revised Manuscript with Track Changes " for detailed insights, particularly Figure 9 data. In this context, the pre-class assessment serves as a proxy for traditional teaching methodologies, whereas the post-class evaluation represents innovative teaching approaches. Consequently, the study does incorporate control data.

Question 2. While quantitative performance is analyzed, student perceptions, satisfaction, and engagement are only briefly mentioned. Please elaborate on these findings as well.

Answer The data from the satisfaction survey has been incorporated into the "Revised Manuscript with Track Changes " on page 23 and 25, within the section titled "Course Satisfaction Survey." This can be seen in Figure 13.

Question 3. The manuscript would benefit from English editing to simplify the writing. Some sections (e.g., tables, explanations of AI functionality) are overly verbose or technically dense for a general readership.

Answer The section titled "Check point" in Table 2 (page 12) has been deleted, and the section labeled "Check name" in Table 3 (page 15) has been modified. Additionally, the feature 2 of "AI " on page 11 has been removed.All of them can be seen from the "Revised Manuscript with Track Changes ".

Question 4. What extent the platform could be generalized to other imaging modalities or clinical areas?

Answer�①The platform is currently integrated with imaging fields such as PET and CT (refer to Annex 2 for specifics). It is projected that within the subsequent five years, it will be extendable to the SPECT/CT imaging field, and within a decade, to the SPECT/MR and other related imaging fields.② Currently, the platform is not only being used by our own institution but has also been expanded to three other medical colleges (see Attachment 2 for details). Within the next five years, it is expected that the platform will cater to undergraduate or standard-training students from 10 medical colleges or affiliated hospitals. Within a decade, the target is to extend this service to 50 medical colleges or affiliated hospitals. This application promotion has been added to the "evised Manuscript with Track Changes " on page 25 in the "Conclusion" section.

These are my comprehensive responses. I eagerly await your feedback.

Best wishes!

Dandan Shang

2025.6.29

---

## [Decision Letter · Decision Letter 1]

3 Sep 2025

Dear Dr. Shang,

We look forward to receiving your revised manuscript.

Kind regards,

Alexandre Bonatto

Academic Editor

PLOS ONE

Journal Requirements:

Reviewers' comments:

Reviewer's Responses to Questions

**Comments to the Author**

Reviewer #3: (No Response)

2. Is the manuscript technically sound, and do the data support the conclusions?

Reviewer #3: Partly

3. Has the statistical analysis been performed appropriately and rigorously?

Reviewer #3: No

4. Have the authors made all data underlying the findings in their manuscript fully available?

Reviewer #3: No

5. Is the manuscript presented in an intelligible fashion and written in standard English?

Reviewer #3: Yes

Reviewer #3: The paper reports several aspects of an online platform aimed to help the training of students on SPECT imaging technology. Half of the paper is devoted to the evaluation of the platform based on its use in the training of more than 500 students. Results are presented in the form of descriptive statistics, and the authors conclude that the platform contributed positively to a better performance of the students in learning the contents of the courses where it has been applied.

I was not a reviewer of the previous version, but I noticed that several questions from the previous reviewers were answered by the authors. However, even after the changes and answers, I think that the paper is not ready for acceptance in its present form.

First of all, the text needs to be restructured and improved.

1. The introduction is too short. The text in the “Platform construction” subsection would be better as part of the introduction. It does not explain the methods used by the authors.

2. I think that Figure 2 needs an improvement and probably a correction. Is the “User experience feedback” the evaluation part? I did not understand the sentence “The formative evaluation of the operator can be relayed to the background, thereby optimizing the system model further.” Is the operator the student? Is the formative evaluation the assessment of the knowledge acquired by the student? These are aspects that should be in the conceptual design framework.

3. Figure 4 is described in the text, referring to numbers for each module. However, the figure itself does not contain the numbers used to identify each module. Also, some expressions are confusing: what is “chairs of landmarks”? Could you use a more specific case for a body weight detector, or are there different such devices? It would be interesting to have images of all modules. What is a ”calling button”?

4. As for Table 2, is “The injection room is filled of medicine” really a “Mode of Injection”?

5. The description starting at line 222 and figure 8 is very confusing. There is no description of Figure 8. Only the list of the stages. Since this is an example, it should be complete and described as a usage scenario. Then, there is a description of the characteristics of the process model, which seems to be out of place. Probably it would be better to move this text to the conceptual design framework section because it is the design rationale of the platform.

The second aspect that deserves profound changes is the “Teaching practice” section, which contains the description of the platform evaluation through its use across time by students.

When assessing the use of a new tool, one usually describes the protocol adopted for composing or inviting the participants, the procedure (i.e., how participants used the tool), the data collected, the data analysis methods, and the results. In the paper, the authors presented the teaching practice in an unstructured way. Please refer to a paper presenting a user study for a better way of reporting such user experiments. Plos One has papers with correct ways of reporting user studies (see, for example, https://journals.plos.org/plosone/article?id=10.1371/journal.pone.0245717)

Moreover, I have some other concerns.

1. If 550 students used the platform, why present only data from 2023-2024?

2. Are the differences between pre-test and post-test scores statistically significant? The authors report only means and standard deviation.

3. As far as I understood, the students followed the three-element classroom approach presented from lines 263-286. The authors mention that the pre-test was applied before the tool was used. So, the students have already engaged in the “pre-class guidance”. On the other hand, the post-test was applied right after the use of the training tool. It is not clear how many times they could perform the simulation during the “interactive learning phase”, and if all students performed the same number of times the simulation tasks.

As a minor comment, please check lines 283-284. What are “Wanqianxing” and “Rain Classroom”?

**Do you want your identity to be public for this peer review?** For information about this choice, including consent withdrawal, please see our Privacy Policy

Reviewer #3: No

---

## [Author Response · Author response to Decision Letter 2]

2 Oct 2025

Dear Editor and Reviewers:

Thank you for your insightful feedback on this article. I will address the concerns of the article sequentially from the perspectives mentioned below.

1 Editor's queries

Question 1. Please include the following items when submitting your revised manuscript:A 'Response to Reviewers'.A 'Revised Manuscript with Track Changes'.A 'Manuscript'.

Answer: The initial draft has been thoroughly revised in accordance with the requested modifications. Please refer to the “Response to Reviewers-2“� " Revised Manuscript with Track Changes-2" and " Manuscript-2" for further details.

Question 2. Answer: I have included a statement of financial disclosure in “cover letter-2”.

Question 3.While revising your submission, please upload your figure files to the Preflight Analysis and Conversion Engine (PACE) digital diagnostic tool, PACE helps ensure that figures meet PLOS requirements.

Answer:I have upload my figures files to the Preflight Analysis and Conversion Engine (PACE) digital diagnostic tool. I have resubmitted the figure files.

Question 4.The PLOS Data policy requires authors to make all data underlying the findings described in their manuscript fully available without restriction.

Answer: All experimental data have been placed in file 2,“Practice data for all students” . Protocols.io has been alerted, and they are currently generating laboratory protocols. If the experimental designs and all associated data are must to be deposited with Protocols.io, these newly generated protocols will supersede file 2 in due course.

2 Questions Raised by Reviewer 3

Question 1. The introduction is too short. The text in the “Platform construction” subsection would be better as part of the introduction. It does not explain the methods used by the authors?

Answer: The content from the subsection titled "Platform construction" has been incorporated into the introduction, and the references suggested by the review experts have been included. This can be found in lines 46-72 of the “Revised Manuscript with Track Changes-2”.

Question 2. I think that Figure 2 needs an improvement and probably a correction. Is the “User experience feedback” the evaluation part? I did not understand the sentence “The formative evaluation of the operator can be relayed to the background, thereby optimizing the system model further.” Is the operator the student? Is the formative evaluation the assessment of the knowledge acquired by the student? These are aspects that should be in the conceptual design framework.

Answer: Figure 2 has been updated�as depicted in Fig 2. "User experience feedback" refers to the evaluation section where operators, which could include students or teachers, participate in a virtual experiment. The data presented in this article is a statistical analysis derived from a population of students within a school setting who have had the opportunity to experience the experiment. The formative assessment focuses on evaluating the students' acquisition of knowledge and their ability to correctly and proficiently operate the experiment. It is also an evaluation of the platform functions, which helps the platform optimize iterations.

Question 3. Figure 4 is described in the text, referring to numbers for each module. However, the figure itself does not contain the numbers used to identify each module. Also, some expressions are confusing: what is “chairs of landmarks”? Could you use a more specific case for a body weight detector, or are there different such devices? It would be interesting to have images of all modules. What is a ”calling button”?.

Answer:. Figure 4 has been updated to incorporate numerical labels for each module, and the module identifiers within the figure have been harmonized with those used in the text. There are no “chairs of landmarks” in the article, only “waiting chairs” and “landmarks”. The weight detector is a standard scale (not a specialised medical one), as can be seen in the following Figure 6. It has been changed to " a weighing scale", as can be seen at line 159 of the “Revised Manuscript with Track Changes-2”. The “mumber calling button” is the button pressed by the doctor to call out the next patient number in the waiting room, as can be seen in Figure 6.The figure 6 has been incorporated at line 169 of the of the “Revised Manuscript with Track Changes-2”. Furthermore, the manuscript incorporates "Figure 8. The waiting room" to facilitate a more intuitive presentation of the content within this platform�as shown at line 174 of the “Revised Manuscript with Track Changes-2”.

Figure 6 The pre-examination preparation room

Question 4. As for Table 2, is “The injection room is filled of medicine” really a “Mode of Injection”?.

Answer: The translation provided is inaccurate. The correct interpretation should be "administering drugs to patients in an injection room," which refers to an "injection mode" of administration that is intravenous. Changes have been made to Table 2, line 184 of the “Revised Manuscript with Track Changes-2”.

Question 5. The description starting at line 222 and figure 8 is very confusing. There is no description of Figure 8. Only the list of the stages. Since this is an example, it should be complete and described as a usage scenario. Then, there is a description of the characteristics of the process model, which seems to be out of place. Probably it would be better to move this text to the conceptual design framework section because it is the design rationale of the platform.

Answer: The operating environment of the original Fig. 8 (now Fig. 10) is elaborated, and the contents in the operating process are elaborated and sorted out, as shown in lines 185 to 214 of the “Revised Manuscript with Track Changes-2”. In addition, this part of the content was added to the conceptual design framework section, as shown in Fig. 2.

Question 6. If 550 students used the platform, why present only data from 2023-2024?

Answer: The SPECT platform undergoes annual optimizations and iterative updates, resulting in minor variations in statistical data collection each year. During its pilot phase, the system solely included training mode data, omitting assessment mode data. By the 2023-2024-1 semester, although data for both training and assessment modes was available, the platform only computed average scores for the training mode, excluding scores from the inaugural training session, thus rendering the data statistics incomplete. It was only in the 2023-2024-2 semester that the platform achieved optimal performance, leading to more comprehensive data statistics. Consequently, the original manuscript chose data from the most recent year as a benchmark. Nevertheless, following expert recommendations, statistical analyses were performed on data from the two latest semesters, specifically 2023-2024-2 and 2024-2025-2. These datasets, derived from the platform's latest iteration and boasting identical data formats, are apt for synchronous analysis. Refer to lines 285-306 and Figure 12 for further details�as in “Revised Manuscript with Track Changes-2”. It is important to note that, due to the lack of a mandatory requirement for the training mode, there were data omissions identified in the records of two students who may have bypassed the training phase and proceeded directly to the assessment phase. As a result, the total amount of data collected from the training mode is slightly less than that gathered from the assessment mode, accounting for the missing data from these two students.

Question 7. Are the differences between pre-test and post-test scores statistically significant? The authors report only means and standard deviation.

Answer: The difference between the pre-class and post-class tests is notable, with the average score of the latter increasing by 12.45%. Furthermore, the post-class test exhibits a larger median and a smaller variance compared to the pre-class test. Please refer to Table 4 for more details. This table has been newly incorporated. see lines 282 of the “Revised Manuscript with Track Changes-2”. Here, in addition, with the optimization of the teaching reform model, only the 2024-2025-2 semester has introduced the ternary classroom of intelligent course teaching. Therefore, the pre-class test and post-class test raw data only contain the data of 129 people in this semester. The reason for this data is also explained in the text, see lines 240 to 242 of the “Revised Manuscript with Track Changes-2”

Question 8. As far as I understood, the students followed the three-element classroom approach presented from lines 263-286 of the “Revised Manuscript with Track Changes-2”. The authors mention that the pre-test was applied before the tool was used. So, the students have already engaged in the “pre-class guidance”. On the other hand, the post-test was applied right after the use of the training tool. It is not clear how many times they could perform the simulation during the “interactive learning phase”, and if all students performed the same number of times the simulation tasks.

Answer: The simulation training is not constrained by a set number of attempts; therefore, during the interaction phase, students may undertake as many training sessions as they require based on their individual needs. This continues until the students feel they have thoroughly mastered the SPECT operation skills and knowledge points. As a result, the frequency of simulation training varies among students. This approach effectively caters to the personalized requirements of the students. This can be seen in lines 117-118 or lines 325 and 380.

Question 9. As a minor comment, please check lines 283-284. What are “Wanqianxing” and “Rain Classroom”?

Answer: “Wanqianxing” and “Rain Classroom” are two distinct teaching tools predominantly utilized for classroom assessments, data analytics, and real-time interactions. This section has been omitted in the latest version,as can be seen in lines 261-265 of the “Revised Manuscript with Track Changes-2”; kindly refer to the most recent draft for precise information.

Furthermore, I have carefully reviewed and revised the unsuitable descriptions present in the text, as outlined in “Revised Manuscript with Track Changes-2”.

These are my comprehensive responses. I eagerly await your feedback.

Best wishes!

Dandan Shang

2025.10.2

---

## [Decision Letter · Decision Letter 2]

19 Nov 2025

Dear Dr. Shang,

**It was extremely challenging to secure reviewers willing to evaluate this manuscript. Hence, as an exception, I decided to proceed based on a single review (because the reviewer is a specialist on the manuscript's subject).**

We look forward to receiving your revised manuscript.

Kind regards,

Alexandre Bonatto

Academic Editor

PLOS ONE

**Journal Requirements:**

Reviewers' comments:

Reviewer's Responses to Questions

**Comments to the Author**

Reviewer #3: (No Response)

2. Is the manuscript technically sound, and do the data support the conclusions?

Reviewer #3: Yes

3. Has the statistical analysis been performed appropriately and rigorously?

Reviewer #3: Yes

4. Have the authors made all data underlying the findings in their manuscript fully available?

Reviewer #3: Yes

5. Is the manuscript presented in an intelligible fashion and written in standard English?

Reviewer #3: Yes

**Reviewer #3:** The paper reports several aspects of an online platform aimed at helping the training of students on SPECT imaging technology. Half of the paper is devoted to the evaluation of the platform based on its use in the training of more than 500 students. Results are presented in the form of descriptive statistics, and the authors conclude that the platform contributed positively to a better performance of the students in learning the contents of the courses where it has been applied.

I was a reviewer of the revised version (not of the original version) and posed comments and suggestions that were mostly followed and answered by the authors. I revised them below and added some extra minor suggestions.

1. The Introduction was improved. The authors kept the text in the “Platform construction” subsection, although it does not explain methods.

I suggest eliminating the title “Principles of Construction and Technical Parameters” and keeping the text as part of “Platform Construction”. This small change will make the section “Platform Construction” really about the techniques used to build it.

An additional correction in the section “Platform construction” is to correct the references in the sentence “Over the past decade, several educators and practitioners, including Jonathan Cooper [9], Tao Shaoneng[10], Oliver A. Meyer [11], and Lawson AP [12], …..” These articles have multiple authors. The authors should just write “Over the past decades, several authors [9,10,11, 12] have ….”

2. Figure 2 has been improved a lot. However, you can simplify the caption by replacing the current one with “Model Design Concept Diagram: Prerequisites, human-computer interaction, and feedback optimization. ”

3. Figure 4 has been improved with the numbers, as has its description in the text. However, there still are “chairs of landmarks” in the figure. I think they are waiting chairs as the authors explained in the response letter. So, the figure needs to be corrected.

4. Table 2 has been corrected as advised.

5. Figure 8 has been thoroughly described in the text as suggested.

As a separate comment, in my previous review, I said that the section “Teaching practice” deserved profound changes. I said, “When assessing the use of some new tool, usually one describes the protocol adopted for composing or inviting the participants, the procedure (i.e., how participants used the tool), the data collected, the data analysis methods, and the results. In the paper, the authors presented the teaching practice in an unstructured way. Please refer to a paper presenting a user study for a better way of reporting such user experiments. Plos One has papers with correct ways of reporting user studies (see, for example, https://journals.plos.org/plosone/article?id=10.1371/journal.pone.0245717)”

The authors made some adjustments in the section but did not add demographic information: for example, the reader does not know basic statistics about age, sex and previous experience with VR. These are common data that one usually adds when reporting experiments.

As for my other concerns (three more questions) and comments, the authors addressed them adequately.

**Do you want your identity to be public for this peer review?** For information about this choice, including consent withdrawal, please see our Privacy Policy

Reviewer #3: No

---

## [Author Response · Author response to Decision Letter 3]

24 Nov 2025

Dear Editor and Reviewers:

Thank you for your insightful feedback on this article. I will address the concerns of the article sequentially from the perspectives mentioned below.

1 Editor's queries

Question 1. Please include the following items when submitting your revised manuscript:A 'Response to Reviewers'.A 'Revised Manuscript with Track Changes'.A 'Manuscript'.

Answer: The initial draft has been thoroughly revised in accordance with the requested modifications. Please refer to the “Response to Reviewers-3“� " Revised Manuscript with Track Changes-3" and " Manuscript-3" for further details.

Question 2. Answer: There added a fund (2024CHYB-48) in my financial disclosure. The statement of financial disclosure can be seen in “cover letter-3”.

Question 3. 1. If the reviewer comments include a recommendation to cite specific previously published works, please review and evaluate these publications to determine whether they are relevant and should be cited. There is no requirement to cite these works unless the editor has indicated otherwise.

Answer: The articles recommended by the reviewers were reviewed, confirming their relevance to this study and suitability for citation. See Ref. 2 under "References", as can be seen at lines 50 and 414.

Question 4. Please review your reference list to ensure that it is complete and correct. If you have cited papers that have been retracted, please include the rationale for doing so in the manuscript text, or remove these references and replace them with relevant current references. Any changes to the reference list should be mentioned in the rebuttal letter that accompanies your revised manuscript. If you need to cite a retracted article, indicate the article’s retracted status in the References list and also include a citation and full reference for the retraction notice.

Answer: The references no longer include citations to retracted or controversial articles . That is, references 3 and 4 were removed, as can be seen at lines 50 and 416–420. The citation numbers in existing references have been adjusted accordingly to maintain consistency with the revised numbering system.

Question 5. If applicable, we recommend that you deposit your laboratory protocols in protocols.io to enhance the reproducibility of your results.

Answer: The laboratory protocol has been deposited in protocols.io. Protocol Integer ID:228888.

Private link: https://www.protocols.io/private/7D8871759FCF11F0A1320A58A9FEAC02.

Or practice data for all students is provided in S2 File.

2 Questions Raised by Reviewer 3

Question 1. The Introduction was improved. The authors kept the text in the “Platform construction” subsection, although it does not explain methods.

Answer: The main technical approaches for constructing the platform were incorporated into the “Introduction”. This can be found in lines 55 to 58 of the “Revised Manuscript with Track Changes-3”. Furthermore, the detailed technical methods are presented in lines 126 to 133 and table 1.

Question 2. I suggest eliminating the title “Principles of Construction and Technical Parameters” and keeping the text as part of “Platform Construction”. This small change will make the section “Platform Construction” really about the techniques used to build it.

Answer: The title "Principles of Construction and Technical Parameters" has been deleted and the text is keeping as part of “Platform Construction”.All changes have been completed. This can be found in lines 113 of the “Revised Manuscript with Track Changes-3”.

Question 3. An additional correction in the section “Platform construction” is to correct the references in the sentence “Over the past decade, several educators and practitioners, including Jonathan Cooper [9], Tao Shaoneng[10], Oliver A. Meyer [11], and Lawson AP [12], …..” These articles have multiple authors. The authors should just write “Over the past decades, several authors [9,10,11, 12] have ….”.

Answer: The changes of this sentence have been completed. This can be found in lines 94 to 96 of the “Revised Manuscript with Track Changes-3”.

Question 4. Figure 2 has been improved a lot. However, you can simplify the caption by replacing the current one with “Model Design Concept Diagram: Prerequisites, human-computer interaction, and feedback optimization. ”

Answer: The caption for Figure 2 has been simplified. This can be found in lines 118-120 and 539-541 of the “Revised Manuscript with Track Changes-3”.

Question 5. Figure 4 has been improved with the numbers, as has its description in the text. However, there still are “chairs of landmarks” in the figure. I think they are waiting chairs as the authors explained in the response letter. So, the figure needs to be corrected.

Answer: The “chairs of landmarks” has changed into “Waiting chairs” and “Landmarks”,as can be seen in figure 4.

Question 6. The authors made some adjustments in the section but did not add demographic information: for example, the reader does not know basic statistics about age, sex and previous experience with VR. These are common data that one usually adds when reporting experiments.

Answer: This paper provides comprehensive details regarding the study participants' demographic characteristics (e.g., age range, gender) and prior VR experience. This can be found in lines 219 to 223 of the “Revised Manuscript with Track Changes-3”.

Furthermore, I have carefully reviewed and revised the unsuitable descriptions present in the text, as outlined in “Revised Manuscript with Track Changes-3”.

These are my comprehensive responses. I eagerly await your feedback.

Best wishes!

Dandan Shang

2025.11.21

---

## [Decision Letter · Decision Letter 3]

14 Jan 2026

Dear Dr. Shang,

We look forward to receiving your revised manuscript.

Kind regards,

Alexandre Bonatto

Academic Editor

PLOS One

Journal Requirements:

Additional Editor Comments:

The reviewer asked a single simple change. Once you implement this change, the paper will be accepted.

Reviewer's Responses to Questions

**Comments to the Author**

Reviewer #3: All comments have been addressed

2. Is the manuscript technically sound, and do the data support the conclusions?

Reviewer #3: Yes

3. Has the statistical analysis been performed appropriately and rigorously?

Reviewer #3: Yes

4. Have the authors made all data underlying the findings in their manuscript fully available?

Reviewer #3: Yes

5. Is the manuscript presented in an intelligible fashion and written in standard English?

Reviewer #3: Yes

Reviewer #3: The authors have addressed all my suggestions, except for Figure 4, which they addressed partially.

They replaced the text "Chairs of landmarks" with "Waiting chairs" in the "Lobby" part of the figure. However, the text "Chairs of landmarks" in the "Waiting room" still needs to be changed.

**Do you want your identity to be public for this peer review?** For information about this choice, including consent withdrawal, please see our Privacy Policy

Reviewer #3: No

---

## [Author Response · Author response to Decision Letter 4]

19 Jan 2026

Dear Editor and Reviewers:

Thank you for your insightful feedback on this article. I will address the concerns of the article sequentially from the perspectives mentioned below.

1 Editor's queries

Question 1. Please include the following items when submitting your revised manuscript:a 'Response to Reviewers'.a 'Revised Manuscript with Track Changes'.a 'Manuscript'.

Answer: The manuscript has been comprehensively revised in response to the recommended modifications.Please refer to the “Response to Reviewers-4“� " Revised Manuscript with Track Changes-4" and " Manuscript-4" for further details.

Question 2. Answer: The financial disclosure statement remains unchanged. The statement of financial disclosure can be seen in “cover letter-4”.

Question 3. 1. If the reviewer comments include a recommendation to cite specific previously published works, please review and evaluate these publications to determine whether they are relevant and should be cited. There is no requirement to cite these works unless the editor has indicated otherwise.

Answer: The articles recommended by the reviewers were reviewed, confirming their relevance to this study and suitability for citation. See Ref. 2 under "References", as can be seen at lines 50 and 410.

Question 4. Please review your reference list to ensure that it is complete and correct. If you have cited papers that have been retracted, please include the rationale for doing so in the manuscript text, or remove these references and replace them with relevant current references. Any changes to the reference list should be mentioned in the rebuttal letter that accompanies your revised manuscript. If you need to cite a retracted article, indicate the article’s retracted status in the References list and also include a citation and full reference for the retraction notice.

Answer: References 13 and 22, which were retracted or controversial, have been removed and replaced with relevant existing literature. This can be found in lines 442-446 and 474-479 of the “Revised Manuscript with Track Changes-4”. The revised reference list no longer includes "withdrawn or controversial references". Additionally, reference 35 was removed due to its duplication of reference 15, with subsequent reference numbers renumbered accordingly. This can be found in lines 382-384 and 516-518 of the “Revised Manuscript with Track Changes-4”. And more, minor revisions have been made to the format of all references in accordance with the template requirements. This can be found in lines 409-542 of the “Revised Manuscript with Track Changes-4”.

Question 5. If applicable, we recommend that you deposit your laboratory protocols in protocols.io to enhance the reproducibility of your results.

Answer: After discussion with the staff of Platform protocols.io, this experimental protocol is deemed unsuitable for implementation on Platform protocols.io. Therefore, this experimental protocol has not been deployed on Platform protocols.io. However, all raw data generated from this experiment have been fully disclosed within the manuscript as “File S2”.

2 Questions Raised by Reviewer 3

Question 1.They replaced the text "Chairs of landmarks" with "Waiting chairs" in the "Lobby" part of the figure. However, the text "Chairs of landmarks" in the "Waiting room" still needs to be changed.

Answer: The “chairs of landmarks” has changed into “Landmarks” and “Waiting chairs”,as can be seen in the "Waiting room" of figure 4. Furthermore, we have further refined Figure 4 by integrating the pathway for nuclear medicine technologists (depicted by yellow arrows in the diagram) and increase explanatory text in the caption of Figure 4: " Blue arrows depict the patient's route, while yellow arrows denote the nuclear medicine technologist's pathway." This can be found in lines 145-146 and 549-550 of the “Revised Manuscript with Track Changes-4”.

These are my comprehensive responses. I eagerly await your feedback.

Best wishes!

Dandan Shang

2026.1.19

---

## [Decision Letter · Decision Letter 4]

26 Jan 2026

An Online Training Platform for SPECT Imaging Technology Utilizing Three-Dimensional Modeling

PONE-D-25-17893R4

Dear Dr. Shang,

We’re pleased to inform you that your manuscript has been judged scientifically suitable for publication and will be formally accepted for publication once it meets all outstanding technical requirements.

Kind regards,

Alexandre Bonatto

Academic Editor

PLOS One

Additional Editor Comments (optional):

Dear Authors,

I am very pleased to inform you that your manuscript has been accepted.

Best regards.

Alexandre Bonatto

Reviewers' comments:

Reviewer's Responses to Questions

**Comments to the Author**

Reviewer #3: (No Response)

2. Is the manuscript technically sound, and do the data support the conclusions?

Reviewer #3: Yes

3. Has the statistical analysis been performed appropriately and rigorously?

Reviewer #3: Yes

4. Have the authors made all data underlying the findings in their manuscript fully available?

Reviewer #3: Yes

5. Is the manuscript presented in an intelligible fashion and written in standard English?

Reviewer #3: Yes

Reviewer #3: The authors have addressed the issue I pointed out while revising version 3 of the manuscript.

They also made changes in the references because they had cited papers that were retracted lately. I'm satisfied with the careful revision.

**Do you want your identity to be public for this peer review?** For information about this choice, including consent withdrawal, please see our Privacy Policy

Reviewer #3: No

---

## [Editor Report · Acceptance letter]

PONE-D-25-17893R4

PLOS One

Dear Dr. Shang,

I'm pleased to inform you that your manuscript has been deemed suitable for publication in PLOS One. Congratulations! Your manuscript is now being handed over to our production team.

Kind regards,

on behalf of

Dr. Alexandre Bonatto

Academic Editor

PLOS One